# *Ganoderma lucidum* Ethanol Extraction Promotes Dextran Sulphate Sodium Induced Colitis Recovery and Modulation in Microbiota

**DOI:** 10.3390/foods11244023

**Published:** 2022-12-13

**Authors:** Miaoyu Li, Leilei Yu, Qixiao Zhai, Bingshu Liu, Jianxin Zhao, Wei Chen, Fengwei Tian

**Affiliations:** 1State Key Laboratory of Food Science and Technology, Jiangnan University, Wuxi 214122, China; 2School of Food Science and Technology, Jiangnan University, Wuxi 214122, China; 3National Engineering Research Center for Functional Food, Jiangnan University, Wuxi 214122, China

**Keywords:** *G. lucidum*, *G. incarnatum*, inflammatory bowel disease, intestinal barrier protection, immune response, gut microbiota

## Abstract

Popular edible mushrooms *Ganoderma lucidum* and *Gloeostereum incarnatum* can improve physical health as a prebiotic and positively alter intestinal microbiota. Our research investigated the prebiotic effects of *Ganoderma lucidum* and *Gloeostereum incarnatum* on colon inflammation through *G. lucidum* water extraction polysaccharides (GLP), *G. incarnatum* water extraction polysaccharides (GIP), *G. lucidum* ethanol extraction (GLE), and *G. incarnatum* ethanol extraction (GIE) administered in mice after 7 days of dextran sulphate sodium (DSS) administration. Among the extracts, GLE showed reduced mortality rates, prevention of weight loss, mitigated colon length shortening, and decreased disease activity indices and histological scores. COX-2, MPO, and iNOS activities and the inflammatory cytokines’ expressions were determined to demonstrate the inhibition inflammation by GLE. Meanwhile, GLE upregulated the levels of MUC2, ZO-1, claudin-3, and occluding to protect the intestinal barrier. Furthermore, GLE modulated the composition of gut microbiota disturbed by DSS, as it decreased the abundance of *Bacteroides*, *Staphylococcus*, and *Escherichia*_*Shigella*, and increased *Turicibacter* and *Bifidobacterium.* Through cell experiment, GLE had a positive influence on adherens junction, tight junction, and TRAF6/MyD88/NF-κB signaling pathways. In conclusion, GLE supplementation promotes DSS-induced colitis recovery by regulating inflammatory cytokines, preserving the intestinal mucosal barrier, positively modulating microbiota changes, and positively influences immune response in TRAF6/MyD88/NF-κB signaling pathways.

## 1. Introduction

Inflammatory bowel disease (IBD), characterized by chronic gut inflammation, affects approximately 11.2 million people worldwide, with the highest incidence in North America and Europe [1]. The clinical manifestations of IBD include abdominal pain, diarrhea, rectal bleeding, constipation, pressing bowel movements, abdominal cramps, tenesmus, vomiting, and nausea [2]. Although studies have revealed the etiology and pathogenesis of IBD, the underlying molecular mechanism is still unknown. Despite such insufficient understanding, some treatments have been proven to be effective against IBD. Effective recovering methods include immunosuppressive drugs, prebiotics, polyphenols, cellulose, and fatty acids [3,4,5,6,7,8,9].

Prebiotics showed alleviative effect on IBD by maintaining the balanced intestinal flora, enhancing the intestinal barrier function, promoting intestinal immune tolerance, interfering with intestinal inflammation, and inhibiting apoptosis of intestinal epithelial cells [10]. Edible mushrooms *Ganoderma lucidum* and *Gloeostereum incarnatum* as prebiotics are used in traditional Chinese medicines to treat various diseases and to promote physical well-being [11,12,13,14]. *G. incarnatum* extracts possess antioxidant, immunomodulatory, anti-inflammatory, antibacterial, and antitumor (anticancer) activities [15,16,17]. *G. lucidum* polysaccharides (GLPs) include (1→3), (1→6)-a/β-glucans, and glycoproteins showed antihypoglycemic, antitumor (anticancer), anti-fatigue, immunomodulatory, antioxidant, antihypolipidemic, anti-inflammatory, and anti-decrepitude (prolonging life) activities [18,19,20,21]. *Ganoderma lucidum* ethanol extracts, including ganoderic acids and saponins, ameliorated lipid metabolic disorders, are hypoglycemic, antioxidant, and modulate the gut microbiota [22,23,24,25].

Edible mushrooms such as *G. lucidum* and *G. incarnatum* are always used in water extraction and ethanol extraction. In our research, we investigated two application methods of *G. lucidum* and *G. incarnatum* as water extraction polysaccharides and ethanol extraction, by whether they had prebiotic impacts in dextran sulfate sodium (DSS)-induced colitis. First, iNOS, MPO and COX-2 activities, integrity of the intestinal barrier, and protection by pro-inflammatory and inflammatory cytokines were evaluated to determine the effects of *G. lucidum* and *G. incarnatum* extracts on DSS-induced colitis. Next, we investigated whether the useful extracts could regulate the composition of the gut microbiota. Finally, through cell experiment, we explored the mechanism by useful extracts in DSS-induced colitis.

## 2. Materials and Methods

### 2.1. Preparation of the G. lucidum Extracts

*G. lucidum* and *G. incarnatum* were obtained in October from Jilin, China and Yunnan, China, respectively. Obtained *G. lucidum* and *G. incarnatum* were dried in 50 °C. *G. lucidum* water extraction polysaccharides (GLP) and *G. incarnatum* water extraction polysaccharides (GIP) were extracted through a 95 °C hot water bath for 30 min three times and then 75% ethanol precipitation. GLE and GIE were extracted through 75% ethanol at 40 °C as previously described [26]. Water extraction polysaccharides of *G. lucidum* and *G. incarnatum* needed removal of residual proteins through the Sevage method, and removed the small molecules through a 3 kDa molecular weight cut-off membrane [27]. All extracts were followed by freeze-drying to obtain GLP, GIP, GLE, and GIE.

### 2.2. DSS-Induced Animal Study

7-week-old male C57BL6/J mice (18 to 22 g) were raised at standard barrier conditions. 80 mice were divided into eight groups and fed sterile water and standard feed. After 7 days, the mice received 3.0% *w/v* DSS drinking water for 7 days, which then was followed by 7 days of normal water. The study protocols were approved by the Ethics Committee of Jiangnan University (JN. No20200710c1720920(186)). An amount of 500 mg/kg GLP, 50 and 100 mg/kg GLE, 500 mg/kg GIP and 100 mg/kg GIE were suspended in a phosphate buffer solution and fed to mice from days 8–14 until the end of the experiments. From day 8 to day 14, morbidity rates, body weight, and pathological characteristics were recorded daily. Disease activity index (DAI) is shown in Table 1 [28], as well as weight changes (%) performance as measured body weight (day 8–14)/body weight (day 0).

### 2.3. Histopathology

The appropriate length colon samples were resolved in 4% paraformaldehyde, embedded in paraffin, sectioned into 4μm thickness, and followed by standard hematoxylin and eosin (H&E) procedures. Histological pathology as crypt damage, severity of inflammation, epithelial erosion, mucosal edema, and goblet cell depletion were recorded as previously described [26].

### 2.4. ELISA Measurement

Frozen colon tissues were suspended with PBS at a 1:9 ratio. After centrifugation at 12,000 rpm for 15 min, the supernatant for activity measurements was collected. MUC2, tight junction (ZO-1 and Occludin), COX-2, iNOS, and MPO were measured using ELISA kits (Nanjing SenBeiJia Biotechnology Co., Ltd., Nanjing, Jiangsu, China) according to the manufacturer’s instructions and as previously reported [29].

### 2.5. Fecal Genomic DNA Extraction and 16S rRNA Sequencing

The FastDNA Spin Kit for Feces (MP Biomedicals, LLC., Irvine, CA, USA) was used to extract Fecal genomic DNA from 0.1 g frozen fecal samples. DNA concentration was measured by a NanoDrop 2000 spectrophotometer (Thermo Scientific, Wilmington, NC, USA). The V3-V4 regions of the bacterial primers were amplified as described [30]. The DNA Gel Extraction Kit (Biomiga, California, USA) was used to purify PCR products, and the Qubit™ dsDNA BR assay kit (Life Invitrogen, Los Angeles, USA) was used to quantify the PCR products. The Illumina MiSeq platform was used to pair end sequenced purified and pooled amplicon libraries.

### 2.6. Caco-2 Cell Experiment

Caco-2 cells were cultured in a DMEM medium (90% MEM basic culture medium, 10% fetal bovine serum, 100 U/mL penicillin, and 100 mg/mL streptomycin), and kept in an incubator at 5% CO_2_, 37 °C, and 95% relative humidity. The concentration of DSS and GLE were obtained through MTT assay [30]. According to the MTT assay results’ selected suitable concentration of DSS and GLE, GLE concentration was at maximum concentration without cytotoxicity and DSS concentration was selected near 50%. Then, seeded cells in 6-well plates at 5.0×10^5^ cells per well were discarded of the supernatant after formation of the epithelial monolayer, washed 3 times by PBS, added to the suitable concentration of DSS and GLE, and cultured for 18 h to collect the cells for qRT-PCR analysis.

### 2.7. Measurement of Cytokines using Reverse-Transcription Quantitative Polymerase Chain Reaction (RT-qPCR)

Total colon tissues (10–20 mg) and cells (2–5 × 10^6^) RNA were extracted using the FastPure^®^ Cell/Tissue Total RNA Isolation Kit (Vazyme Biotech Co., Ltd, Nanjing, China), transcribed to cDNA using the HiScript^®^ Ⅲ All-in-one RT SuperMix Perfect for qPCR (Vazyme Biotech Co., Ltd, Nanjing, China), and qRT-PCR analysis was carried out using the Taq Pro Universal SYBR qPCR Master Mix (Vazyme Biotech Co., Ltd, Nanjing, China) [31]. The 2^−∆∆Ct^ method was used to perform quantification, and the quantification was a standardized expression of GAPDH or β-actin and expressed as a fold change compared to the control group. The sequences of all primers used for RT-qPCR are listed in Table 2.

### 2.8. Statistical Analysis

SPSS software 20 and GraphPad Prism 7 were carried out for statistical analysis. QIIME 2 was used for Microbiota-relevant analysis; the linear discriminant analysis (LDA) and LDA effect size (LEfSe) method were applied to analyze the predominance of bacterial communities between groups, based on the Kruskal–Wallis rank-sum test (*p* < 0.05) to determine significantly different abundances and the LDA score(log10) = 4.0 as the cut-off value.

## 3. Results

### 3.1. Influence of G. lucidum and G. incarnatum Extract Administration on the Recovery of DSS-Induced Colitis

DSS-induced colitis in the mouse model was used to evaluate whether *G. lucidum* and *G. incarnatum* extracts had an obviously prebiotic effect on IBD. DSS-induced colitis solution is shown in Figure 1A. After analysis, we found that GLE administration at 50 and 100 mg/kg showed better prebiotic effects in promoting recovery from colitis, displayed as significantly decreased weight loss (Figure 1B), reduced mortality (Figure 1C), reduced weight loss (Figure 1B), significantly decreased colon shortening (Figure 1D), and lower DAI scores were observed (Figure 1E). Compared with the recovery group, GLP administration in 500 mg/kg didn’t show effectiveness in promoting recovery in decreased weight loss (Figure 1B), reduced mortality (Figure 1C), reduced weight loss (Figure 1B), decreased colon shortening (Figure 1D), and lower DAI scores (Figure 1E). Compared with GLP administration in 500 mg/kg, 500 mg/kg GIP, and 100 mg/kg, GIE administration promoted weight gain while it showed no difference in reduced mortality (Figure 1C), reduced weight loss (Figure 1B), decreased colon shortening (Figure 1D), and lower DAI scores (Figure 1E).

### 3.2. Influence of G. lucidum and G. incarnatum Extract Administration on DSS-Induced Colonic Tissue Damage and Regulation of Inflammatory Enzymes

To further evaluate its prebiotic effect of GLP, GLE, GIP, and GIE against DSS-induced colitis in mice, histological analyses were performed by staining with hematoxylin and eosin (H&E). In the recovery, GLP, GIP, and GIE groups, locally saw more epithelial cells and incomplete structures of the mucosal layer, connective tissue hyperplasia with a small amount of lymphocytic infiltration, and more focal lymphocytic infiltration in the lamina propria (Figure 2A). Low histological scores, epithelial erosion scores, goblet cell depletion scores, crypt damage scores, mucosal edema scores, and inflammatory infiltration scores were shown in the 50 and 100 mg/kg GLE groups, and the lower scores were in 100 mg/kg GLE (*p* < 0.05) (Figure 2B–G). In the 50 and 100 mg/kg GLE groups, the mucosal layer was structurally intact, with closely arranged epithelial cells and abundant glands in the lamina propria, and a small local focal infiltration of lymphocytes (Figure 2A). Compared with the recovery group, the GLP and GIE groups showed no significant difference in the histological score, and the epithelial erosion, goblet cell depletion, crypt damage scores, mucosal edema, and inflammatory infiltration showed no significant difference (Figure 2B–G). The GIP-treated exhibited a lower mucosal edema score than the recovery group, and other scores revealed a consistent trend (Figure 2B–G).

Activities of COX-2, iNOS, and MPO were analyzed to evaluate the effects of *G. lucidum* and *G. incarnatum* extracts on colonic inflammatory enzymes. No significant difference was seen in MPO concentrations on control, DSS, recover, GLP, GLE, GIP, and GIE groups (Figure 3B). Compared with the recovery group, GLP administration at 500 mg/Kg, GLE administration at 50 and 100 mg/kg significantly decreased the concentrations of iNOS and COX-2 (Figure 3A,C). The concentrations of iNOS and COX-2 in GLE administration were lower than in GLP administration (Figure 3A,C). Compared with the recovery group, GIE groups showed no difference in the concentrations of iNOS and COX-2 (Figure 3A,C).

### 3.3. Effects of G. lucidum Extract Administration on the Regulation of Inflammatory Cytokines

The expression of TNF-α, IL-1β, IL-17, IL-6, PPAR-γ, and IL-10 were assayed to assess the effects of *G. lucidum* and *G. incarnatum* extracts on the modulation of inflammatory cytokines. Demonstrated in Figure 4, compared with the recovery group, 500 mg/kg GLP, 500 mg/kg GIP, and 100 mg/kg GIE administration showed no significant difference in the expression of pro-inflammatory cytokines (including IL-6, IL-1β, IL-17, and TNF-α) and anti-inflammatory cytokines (including IL-10 and PPAR-γ). GLE at 50 and 100 mg/kg displayed a significant decrease expression in pro-inflammatory cytokines, including IL-6, IL-1β, IL-17, and TNF-α, and a significant increase expression in anti-inflammatory cytokines, including IL-10 and PPAR-γ (*p* < 0.05).

### 3.4. Effects of G. lucidum Extract Administration on the Protection of Intestinal Barrier

The concentration of mucin 2 (MUC2), claudin-3, and occludin ZO-1 were measured to analyze the effects of *G. lucidum* and *G. incarnatum* extracts for intestinal barrier protection. Firstly, DSS indeed destroyed the intestinal barrier and significantly decreased the concentrations of MUC2, ZO-1, Claudin-3, and Occludin (Figure 5). During the recovery period, the intestinal barrier showed repairing effects with increased concentrations of ZO-1 and Claudin-3, while there were no differences in concentrations of MUC2 and Occludin (Figure 5). The concentrations of MUC2, ZO-1, Claudin-3, and Occludin in GLE administration showed no significant difference with the control group, indicating that GLE administration promoted the reparation of the intestinal barrier and showed no difference with the control group (Figure 5). The concentrations of MUC2, ZO-1, Claudin-3, and Occludin in GLP, GIP, and GIE administration showed no different in recovery, and demonstrated that GLP, GIP, and GIE could not promote the recovery in the intestinal barrier (Figure 5).

### 3.5. Intestinal Microbiota Modulation by GLE Administration

We analyzed the gut microbiota structure at different levels by GLE administration. *Deferribacteres*, *Bacteroidetes*, *Actinobacteria*, *Verrucomicrobia*, *Proteobacteria*, and *Firmicutes* represented dominant bacteria of all groups at the phylum level (Figure 6). Shown from the Shannon index, the GLE-treated group had a higher alpha multiplicity compared with the recovery group (*p* < 0.05), while there was no significant difference in alpha diversity between the recovery group and GLE-treated group (Figure 6A). From the Principal coordinate analysis (PCoA), there was community structures between the recovery group and GLE-treated groups, indicating a separation in compositional structures (Figure 6B). LEfSe was used to analyze the gut microbiota diversity; with LDA set as 4.0, we could see dominant communities of ten, seven, six, and ten taxa were found in the model, recovery, low-dose GLE-treated, and high-dose GLE-treated groups, respectively. Additionally, compared with the model group, the Firmicutes/Bacteroidetes (F/B) ratio was significantly decreased, while there were no differences in the group of control, recovery, low-dose-GLE, and high-dose-GLE (Figure 6D). The heat map at the genus level is displayed, compared with the recovery group. 100 mg/kg GLE significantly increased the abundance of *Turicibacter* and *Bifidobacterium* and decreased the abundance of *Bacteroides*, *Staphylococcus*, and *Romboutsia*; 50 mg/kg GLE significantly increased the abundance of *Enterococcus* and *Parabacteroides* and decreased the abundance of *Staphylococcus* and *Romboutsia* (Figure 6E).

### 3.6. GLE Improved Intestinal Barrier Function and Inhibited TRAF6/MyD88/NF-κB Signaling

To analyze the regulation mechanism of GLE by DSS induced colitis, we used Caco-2 cells to investigate the signaling of the intestinal barrier function and inflammation. From the MTT analysis, the appropriate concentration of DSS was 3% DSS, where cell viability was near 50%, and the maximum safe concentration of GLE was 0.25 mg/mL, where cell viability was nearly 100% (Figure 7A,B). From the analysis of the results, with 3% DSS induced Caco-2 cells regulated by 0.25 mg/mL GLE, GLE had a positive effect on the intestinal barrier function including the tight junction and adherens junction, as well as inhibited the TRAF6/MyD88/NF-κB signaling pathway (Figure 7C–F). Specifically, 0.25 mg/mL GLE could significantly increase the expression of Claudin-3, Claudin-1, ZO-1, Occludin, cdc42, Par3, RhoA, IRSp53, and Arp3, significantly decrease the mRNA expression levels of NFκB, TRAF6, JNK, MyD88, and IRF-7, and had no significant difference in the expression of TBK1, IRF-3, and MEKK.

## 4. Discussion

*G. lucidum* ethanol extract was enriched with triterpenoids, including ganoderic acids and saponins, which has wide pharmacological effects, such as anti-obesity, increased immunity, and anti-inflammatory activities [23,32,33]. In addition, our study certificated *G. lucidum* ethanol extract, showing a prebiotic effect on DSS-induced colitis. Although Guo et al. demonstrated that the application of GLP prevented DSS-induced colitis in mice by enhancing the intestinal barrier function, increasing the production of SCFAs, and regulating the intestinal microbiota [34], our research of GLP did not display promoting effects in the recovery from colitis.

*G. lucidum* ethanol extract increased immunity during the recovery period. *G. lucidum* ethanol extract administration significantly reduced iNOS and COX-2 activities. High levels of COX-2 are associated with gastroenteritis and unregulated iNOS is frequently noted in chronic inflammation, and these molecules are classical biomarkers for assessing the level of inflammatory response in the colon [35,36,37]. Also, *G. lucidum* ethanol extract administration decreased the expression of inflammatory mediators IL-1β, IL-6, IL-17, and TNF-α in DSS-induced colitis. The levels of inflammatory factors contribute to the initiation, development, and progression of DSS-induced colitis [38]. Overproduction of pro-inflammatory cytokines, including TNF-α, IL-1β, IL-17, and IL-6, causes severe tissue damage and enhances the inflammatory response [39,40,41,42]. Meanwhile, after GLE administration, the levels of anti-inflammatory cytokines (IL-10 and PPAR-γ) were consistent with those in the control group. IL-10 and PPAR-γ are immune-modulating cytokines with anti-inflammatory activity, and they have shown the capability to reduce clinical symptoms in patients with IBD [43]. Edible mushrooms could enhance immune responses via MAPK and NFκB signal pathways [44], and GLE administration immune responses were in TRAF6/MyD88/NF-κB signaling pathways.

Intestinal inflammation in IBD can disturb the epithelial barrier integrity, causing increased permeability and infiltration of pathogens [45]. GLE administration promoted the recovery of the intestinal barrier through promoting the expression of the tight junction and adherens junction. Therefore, after GLE administration, the concentrations of MUC2, ZO-1, claudin-3, and occludin were similar with the control group. The mucosal barrier can protect the intestinal mucosa from damage and stop exogenous substances from destroying the intestinal tissues [46]. Tight junction proteins (ZO-1, claudins, and occludin) governed the intestinal barrier and play a critical role in IBD conservation [47].

The gut microbiome characterized by IBD patients exhibits decreased profitable metabolites (as butyric acid), enrichment of the phylum Aspergillus (as *Escherichia coli*), and depletion of strictly anaerobic bacteria [48]. In our study, 16S rRNA sequencing was used to examine the prospective alterations in microbial composition and diversity. GLE-treated mice showed significant clustering separation from DSS-induced mice using PCoA, which suggests that GLE treatment noticeably altered the biological structure. Through LEfSe analysis, *Escherichia*_*Shigella* showed relative enrichment in DSS-induced mice. Gram-negative bacterium *Escherichia*_*Shigella* leads to diarrheal diseases worldwide [49]. Its levels significantly increase in Crohn’s disease remission [21]. In our results, the relative abundance of *Escherichia*_*Shigella* was significantly higher in DSS-induced colitis mice, and during the recovery period, it was reduced. After GLE administration, the relative abundance of *Escherichia*_*Shigella* was lower compared to that in the recovery group, and the lowering was correlated with the dose of GLE. *Bacteroides* and *Staphylococcus* displayed a relative enrichment in the recovery group, and the heatmap showed that *Staphylococcus* (a gram-positive bacterium) was more abundant during the recovery period; after GLE administration, the abundance of *Staphylococcus* decreased. The abundance of *Bacteroides* is negatively correlated with the extent of tight junction proteins (ZO-1, occluding, and claudin-1), and positively associated with levels of pro-inflammatory cytokines (IL-1β, TNF-α, and IL-6) [49]. There was a significant increase of *Bacteroides* abundance during the recovery period, and high-dose GLE administration could reduce the abundance of *Bacteroides*. *Parabacteroides* displayed relative enrichment in low-dose GLE-treated mice, which was associated with amino acid metabolism [50]. *Turicibacter* and *Bifidobacterium* showed relative enrichment in high-dose GLE-treated mice. *Turicibacter* is a genus of anaerobic gram-positive bacteria, and reduced *Turicibacter* abundance is shown in obesity and irritable bowel syndrome [51,52,53]. Moreover, with *Turicibacter* and *Bifidobacteria* as probiotics, the former can influence gastrointestinal motor patterns and improve the production of SCFAs, while the latter can interact with the host and play a positive regulation role on the immune system [54,55]. *Bifidobacteria* can alleviate DSS-induced colitis by regulation of the intestinal microbiota and maintain the mucosal barrier [35,56]. In addition, the abundance of *Lactobacillus* increased significantly following GLE administration, whatever the dose. The *Lactobacillus* genera have been previously studied to mediate host immunity by regulating T lymphocytes, macrophages, and natural killer (NK) cells [57,58]. In addition, many studies certificated that *Lactobacillus* could relieve DSS-induced colitis by modulating gut microbiota composition and immune response [30,59,60].

## 5. Conclusions

In conclusion, we evaluated the prebiotic effects of *G. lucidum* and *G. incarnatum*’s different consumption ways (GLP, GLE, GIP, and GIE) on DSS-induced colitis. GLE demonstrated effective effects in promoting recovery from colitis as it prevented weight loss, reduced mortality, relieved colonic shortening, and reduced DAI and histological scores. In contrast, GLP, GIP, and GIE did not have a positive effect on relieving colitis. Recovery-promoting effects of GLE are mainly achieved by increasing immunity and protecting the intestinal barrier. In addition, GLE administration altered the gut microbiota composition, significantly increased the abundance of *Turicibacter*, *Bifidobacterium*, and *Parabacteroides*, and decreased the abundance of *Escherichia*_*Shigella*, *Bacteroides*, and *Staphylococcus*.

## Figures and Tables

**Figure 1 foods-11-04023-f001:**
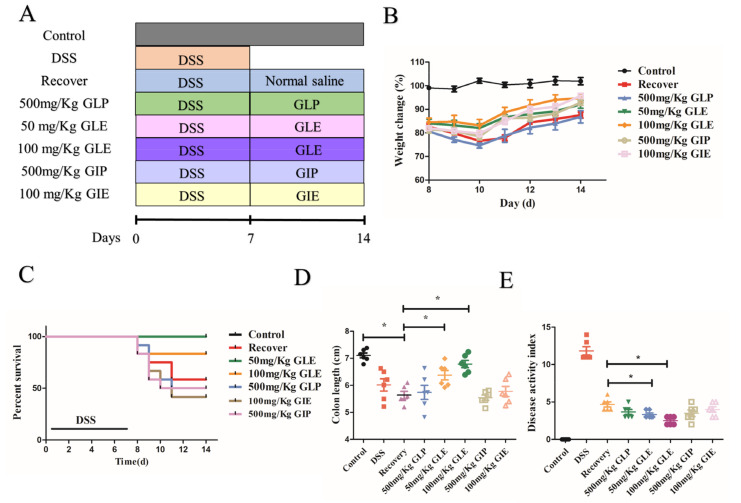
The prebiotic effect of *G. lucidum* and *G. incarnatum* extract in DSS-induced colitis. (**A**) Schematic diagram of the experimental schedule. (**B**) Body weight change (%). (**C**) Survival rate. (**D**) Colon length (cm). (**E**) Disease activity index. *n* = 8 mice per group, the mean values ± SD are presented, * *p* < 0.05.

**Figure 2 foods-11-04023-f002:**
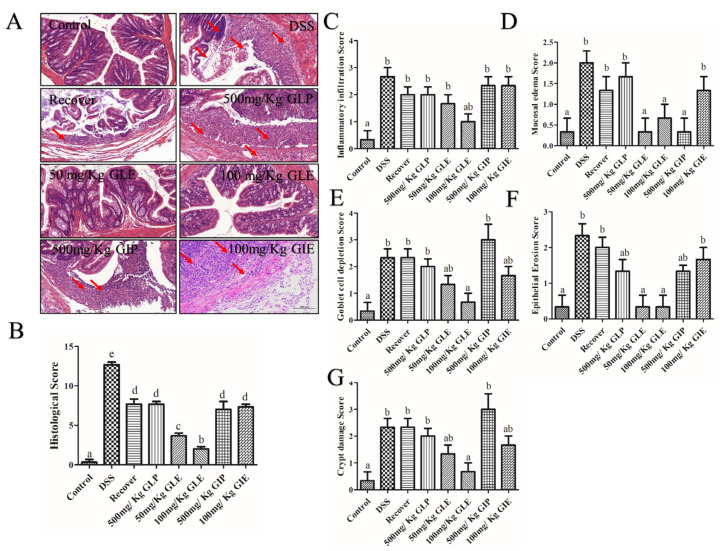
Recovery of histological injury in DSS-induced colitis. (**A**) H&E staining (20×). (**B**) Histology score. (**C**) Epithelial erosion score. (**D**) Crypt damage score. (**E**) Inflammatory infiltration score. (**F**) Goblet cell depletion score. (**G**) Mucosal edema score. *n* = 6 mice per group, the mean values ± SD are presented. Bars with different letters were considered significant at *p* < 0.05.

**Figure 3 foods-11-04023-f003:**
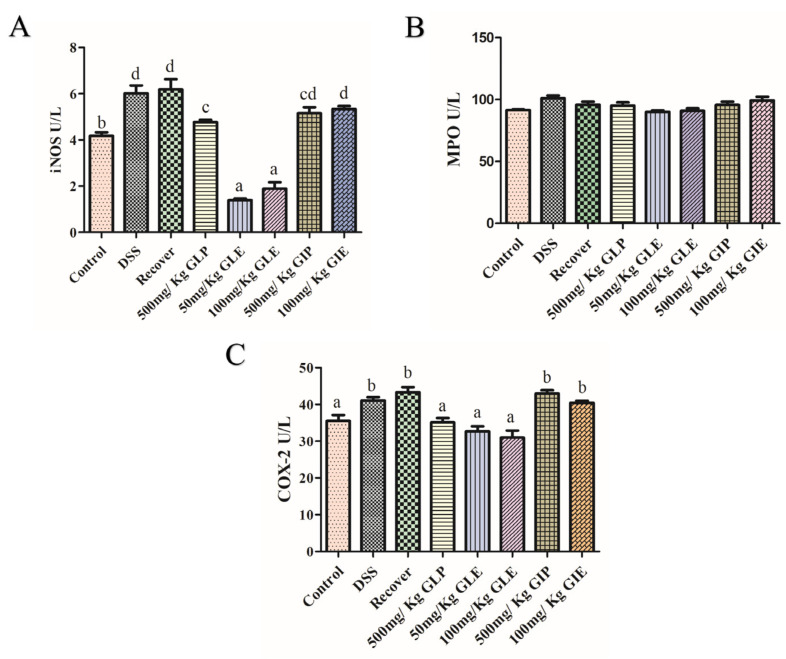
Prebiotic effect of *G. lucidum* and *G. incarnatum* extract on enzymes activities in DSS-induced colitis. (**A**) iNOS, (**B**) MPO, and (**C**) COX-2. *n* = 6 mice per group, the mean values ± SD are presented. Bars with different letters were considered significant at *p* < 0.05.

**Figure 4 foods-11-04023-f004:**
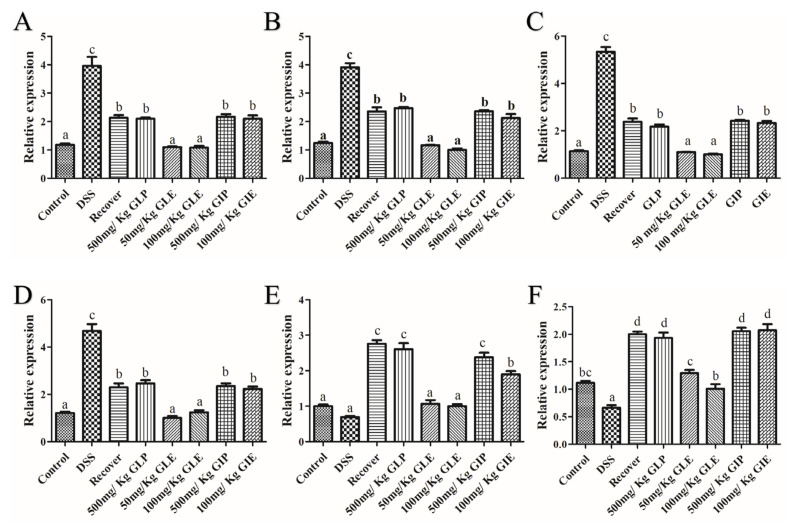
Effects of *G. lucidum* extract administration on the regulation of inflammatory cytokines. (**A**) TNF-α, (**B**) IL-1β, (**C**) IL-17, (**D**) IL-6, (**E**) PPAR-γ, (**F**) IL-10. *n* = 6 mice per group, the mean values ± SD are presented. Bars with different letters were considered significant at *p* < 0.05.

**Figure 5 foods-11-04023-f005:**
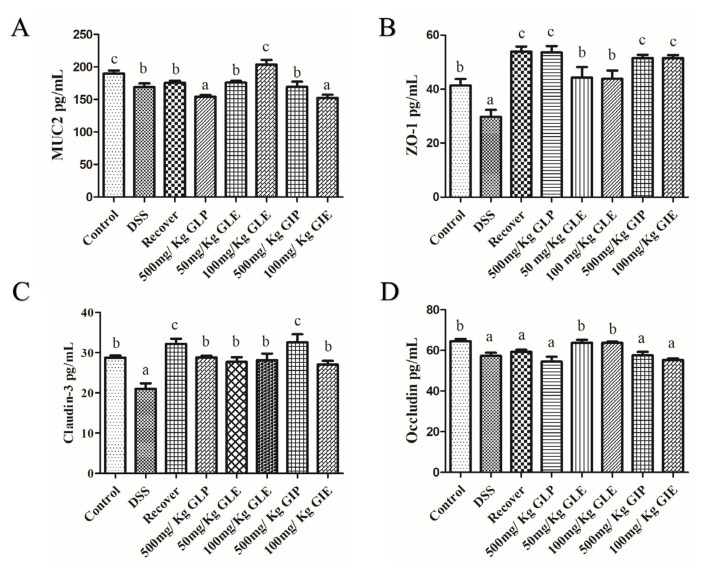
Protection of the intestinal barrier by GLP, GLE, GIP, and GIE administration. (**A**) ZO-1, (**B**) claudin-3, (**C**) occludin, and (**D**) MUC2. *n* = 6 mice per group, the mean values ± SD are presented. Bars with different letters were considered significant at *p* < 0.05.

**Figure 6 foods-11-04023-f006:**
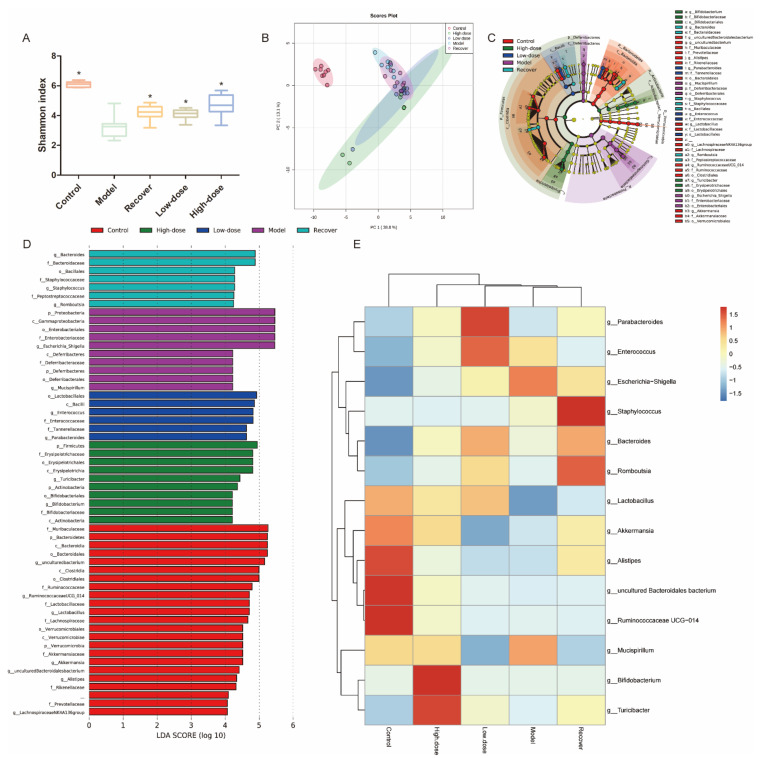
Influence of microbiota modulation by GLE added in DSS-induced colitis. (**A**) Alpha diversity boxplot (Shannon) in colon. (**B**) PCoA using unweighted-UniFrac of beta diversity in colon of GLE group. (**C**) Taxonomic cladogram from LEfSe in colon intestine. (**D**) Differences among the control, high-dose, low-dose, model, and recovery groups in colon intestine, LDA score > 4. (**E**) Heatmap at the genus level. * *p* < 0.05.

**Figure 7 foods-11-04023-f007:**
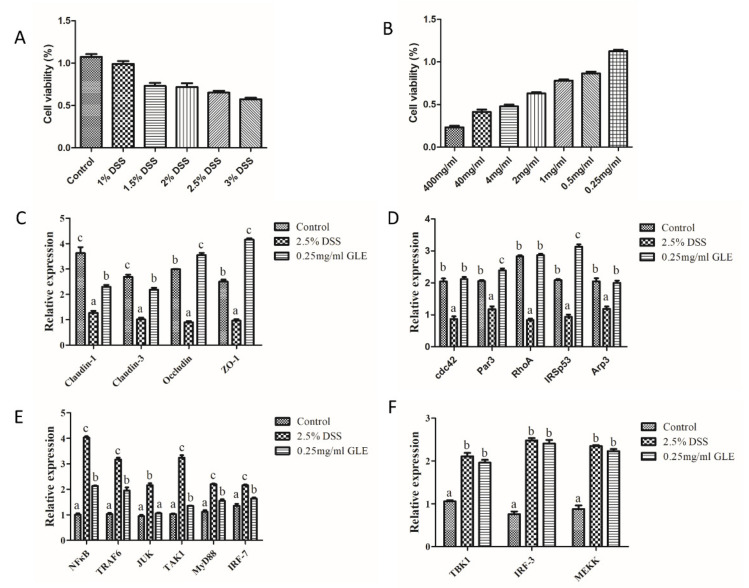
GLE inhibited DSS-induced intestinal barrier function and inflammation. (**A**) MTT in DSS, (**B**) MTT in GLE, (**C**) Tight junction signaling, (**D**) Adherens junction signaling, (**E**) NFκB, MyD88 and JUK signaling, and (**F**) TBK1 and MEKK signaling. mean values ± SEM are presented (*n* = 6), Bars with different letters were considered significant at *p*<0.05.

**Table 1 foods-11-04023-t001:** Scoring system for Disease Activity Index (DAI).

Score	Weight Loss	Stool Consistency	Blood Stool
0	No loss	Normal	No blood
1	1–5%	Loose stool	
2	5–10%	Watery diarrhea	Presence of blood
3	10–20%	Slimy diarrhea, little blood	
4	>20%	Severe watery diarrhea with blood	Gross bleeding

**Table 2 foods-11-04023-t002:** Primer sequences used for qPCR.

Gene	Sequence (5’ to 3’)	
	Forward	Reverse
GAPDH	GACAAGCTTCCCGTTCTCAG	GAGTCAACGGATTTGGTCGT
IL-6	ACCAGAGGAAATTTTCAATAGGC	TGATGCACTTGCAGAAAACA
TNF-α	TGCCTATGTCTCAGCCTCTTC	GGTCTGGGCCATAGAACTGA
IL-1β	ACCTTCCAGGATGAGGACATGA	CTAATGGGAACGTCACAC ACCA
IL-17	CTCCAGAAGGCCCTCAGACTAC	GGGTCTTCATTGCGGTGG
PPAR-γ	CTGCTCAAGTATGGTGTCCATGA	TGAGATGAGGACTCCATCTTT ATTCA
IL-10	GCTCTTACTGACTGGCATGAG	CGCAGCTCTAGGAGCATGTG
β-actin	CCTTCCCTCCTCAGATCATTGC	ATACTCCTGCTTGCTGATCCAC
Claudin-1	TCTATGACCCTATGACCCCAGT	TCTGGGAAATGATGGCACTAGC
Claudin-3	CGAGAAGAAGTACACGGCCAC	GTCTGTCCCTTAGACGTAGTCC
Occludin	CATTAACTTCGCCTGTGGATGAC	TCTCTTTGACCTTCCTGCTCTTC
ZO-1	AGTACCAGAAATACCTGACGGTG	CTTGGCTGACACTAGAAGTAGCA
MYD88	TCGAAAAGAGGTTGGCTAGAAGG	CTTGCTCTGCAGGTAATCATCAG
IRF-7	CCCATCTTCGACTTCAGAGTCTT	CGAAGCCCAGGTAGATGGTATAG
NFκB	AGCTTCAGAATGGCAGAAGATGA	CAGTGCCATCTGTGGTTGAAATA
TRAF6	TTGCTCTTATGGATTGTCCCCAA	GACAGTTCTGGTCATGGATCTCT
TAK1	GAGATCAAGAGGGTGATGCAGAT	CGAGTGATAAGCACATTAGCAGC
JUK	GCCACAAAATCCTCTTTCCAGG	AGGACATCAGGGAAGAGTTTCTC
MEKK	TCACGAAGGAATCAAGAGAGCAA	AAAATAAGCAGCCAACGAGTTCC
TBK1	ACAGATTTTGGTGCAGCTAGAGA	TACCCCAATGCTCCAAAGATCAA
IRF-3	CAAAGAAGGGTTGCGTTTAGCA	ACTCCAGATATTGCACCAGAAGG
Arp3	GCTGCATGAAAATTCAGTTCGTC	GCCACAGAGAAGATTCTTAGCCT
RhoA	TCCGGAAGAAACTGGTGATTGTT	TCAGGCGATCATAATCTTCCACA
IRSP53	CGACTCCTACTCCAACACACTC	CAGAGTCTTGTTCTCGGTGGTG
cdc42	CAGAAGCCTATCACTCCAGAGAC	GCAGCCAATATTGCTTCGTCAAA
Par3	CTAATTGGCCTCTCCACTTCTGT	TCCCATCCTCATCCTTCCTGTC

## Data Availability

The data showed in this study are contained within the article.

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
