# Peer review of "Ganoderma lucidum Ethanol Extraction Promotes Dextran Sulphate Sodium Induced Colitis Recovery and Modulation in Microbiota"

_foods, 2022, doi:10.3390/foods11244023_

Round 1

Reviewer 1 Report

The manuscript “Appropriate consumption of Ganoderma lucidum promote recovery from DSS-induced colitis through intestinal mucosal barrier, signaling pathway and positive modulation in microbiota”. Any natural things with nutritional and medicinal properties are highly appreciable as a substitution for chemical ones. The study is good and the findings are satisfactory. However, a few points need to address.

1.       Write the full name of DSS while appearing first in the MS.

2.       The methodology is moderate for example ELISA measurement, the author needs to write the methodology in detail so that it can explain the process and help others to follow this.

3.       Also, add in the methodology how the author calculated the weight changes (%) (figure 1)

4.       The figure legend GLP, GIP, and GIE is not representing the amount. Add these values in figure 1.

5.       How the author calculated the Disease activity index (figure 1) mentions this.

6.       In figure 2 legend author needs to mention the statistical difference i.e p<0.05 compare wiTh what, as it seems it is not the ANOVA analysis, If it is a T-test author much add the values compared with what control or DSS or Recover. Follow the same for figures 3 and 4 also.

7.       Statistics in Figures 5 and 7 ANOVA. Why author perform two different statistical tests? Explain

8.       Figure 8 B, PCA resolution is very poor. Improve the resolution of the figure for better clarity.

9.       Over the MS need for English checking.

Author Response

Response to editor and reviewers’ comments

Dear Editors and Reviewers,

Thank you for the comments and helpful suggestions, which are very helpful for revising and improving our paper, as well as the important guiding significance to our further research. We have studied comments carefully and have revised this manuscript accordingly. Changes are highlighted in yellow, and responses to their specific comments are detailed below.

Sincerely yours,

Dr. Fengwei Tian

State Key Laboratory of Food Science and Technology

Jiangnan University

Wuxi 214122, P. R. China

fwtian@jiangnan.edu.cn

Q1:  Write the full name of DSS while appearing first in the MS.

Response: Thank you for your reminder. We had corrected in line 23 as “dextran sulphate sodium (DSS)”.

Q2: The methodology is moderate for example ELISA measurement; the author needs to write the methodology in detail so that it can explain the process and help others to follow this.

Response: Thank you for your comments. We supplemented the material and methods as season of collection of medical plants, treatment, and the reference of remove of proteins in line 74-78, as “G. lucidum and G. incarnatum were obtained in October from the Jilin, China and the Yunnan, China, respectively. Obtained G. lucidum and G. incarnatum were dried in 50℃. G. lucidum water extraction polysaccharides (GLP) and G. incarnatum water extraction polysaccharides (GIP) were extracted through 95℃ hot water bath for 30 minutes 3 times and then 75% ethanol precipitation” and the reference [27] in line 82, ELISA measurement in line 104-108, as “Frozen colon tissues were suspended with PBS at 1:9 ratio. After centrifugation at 12,000 rpm 15 min, collected the supernatant for activity measurements. MUC2, tight junction (ZO-1 and Occludin), COX-2, iNOS, and MPO were measured using ELISA kits (Nanjing SenBeiJia Biotechnology Co., Ltd., Nanjing, Jiangsu, P. R. China) according to the manufacturer’s instructions and as previously reported [29]”, and the reference [31] in reverse-transcription quantitative polymerase chain reaction in line 133.

Q3: Also, add in the methodology how the author calculated the weight changes (%) (figure 1)

Response: Thank you for your reminder. Weight changes (%) performance as measured body weight (day 8-14)/ body weight (day 0), shown in line 93-94 as “weight changes (%) performance as measured body weight (day 8-14)/ body weight (day 0)”.

Q4: The figure legend GLP, GIP, and GIE is not representing the amount. Add these values in figure 1.

Response: Thank you for your reminder. We had completed the legend name of the figure 1.

Q5: How the author calculated the Disease activity index (figure 1) mentions this.

Response: Thank you for your comments. We had completed the scoring system for Disease Activity Index (DAI) in Table 1.

 Table 1. Scoring system for Disease Activity Index (DAI)

Score

Weight loss

Stool consistency

Blood stool

0

No loss

Normal

No blood

1

1-5%

Loose stool

2

5-10%

Watery diarrhea

Presence of blood

3

10-20%

Slimy diarrhea, little blood

4

>20%

Severe watery diarrhea with blood

Gross bleeding

Q6: In figure 2 legend author needs to mention the statistical difference i.e p<0.05 compare with what, as it seems it is not the ANOVA analysis, If it is a T-test author much add the values compared with what control or DSS or Recover. Follow the same for figures 3 and 4 also.

Response: Thank you for your comments. We re-analyzed the data using the ANOVA analysis, shown in figure 2, figures 3 and 4.

Figure 2

Figure 3

Figure 4

Q7: Statistics in Figures 5 and 7 ANOVA. Why author perform two different statistical tests? Explain

Response: Thank you for your comments. We re-analyzed the data all using the ANOVA analysis.

Figures 5

Figures 7

Q8: Figure 6B, PCA resolution is very poor. Improve the resolution of the figure for better clarity.

Response: Thank you for your comments. We used a clearer picture of PCA in Figure 6B.

Figure 6

Q9: Over the MS need for English checking.

Response: Thank you for your advice. We checked grammatical errors in full text to avoid this error, shown in line 49-61, line 148-161, line 169-185, line 288-291, line 293-295, line 309-311, line 312-316, line 324-326, line 337-341, line359-362 and line 363-364.

Reviewer 2 Report

foods-1926997-peer-review-v1

The present work come with a promising title, however, even if the authors performed a series of interesting experiments, way of presentation was very poor and manuscript needs serious reorganizations and updating. Unfortunately, in present way, the manuscript is unacceptable, since is wrong exploring some scientific terms, manuscript missing appropriate structure, discussion needs extensive revision, etc.

In my opinion, title is very long and can be presented in better, more compact way.

In the affiliation section, authors need to use say style of presenting their addresses. Please, decide if you can call your country PR China or China. Personally, I would like to follow names according to the United Nation List.

In order to be called probiotic, a microorganism supposed to be provided as life and in certain numbers to the host where can exhibits his beneficial properties. Associated with the definition, edible mushroom will be difficult to be defined as probiotics. Maybe different, more appropriate way of calling them need to be choice by the authors. If authors using different extracts from these mushrooms, then this extract can be considered as prebiotics, but this is different term, referring to the nondigestive organic structure that can be associated with the probiotics (microorganisms) growth. However, this need to be well defined by the authors.

Ln14-18: All information presented on this section is showing well, that authors exploring PREBIOTIC role of the GLP, GIP, GLE, GIE on the specific bacterial species presented in the GIT or direct effect of those extracts and consequence effecting the colon inflammation. However, this is not a PROBIOTIC effect. Please, check what is probiotic, prebiotic.

Ln42-45: No relevance with the followed research presented in the manuscript.

Ln45: Provide references about polyphenols from mushrooms.

Ln45: Mushrooms are not probiotics. They can serve as source of prebiotics.

Ln58-62 can be combined and avoid repetitions.

Ln62: Again, in this and several other places word probiotic was wrong used. In some cases, can be replaced by prebiotic.

Introduction need to be prepared according to the specific meaning of the terms. As well, information in the introduction needs to be clearlly associated with the feature presented information explored by the article. Please, reorganize and present Introduction in better way.

Ln81: Add country, presume that this is China.

Ln78-86: Process of preparation of the extract need to be presented better with sufficient details in order experiments to be repeated. Moreover, it is well known that season of collection of medical plants, including mushrooms is important for the presence of some components. Thus, period of collection of mushrooms and their keeping/treatment as well need to be provided. Way of remove of proteins, as well need to be described, and appropriate reference to be provided.

Most of the material and methods were described in a very poor way, missing details, missing references.  

Results are presented more-or-less in a good shape, however, most probably based on unexperienced position of the authors, in several places wrong terminology was used. Maybe authors can look for a help from more experience colleagues to help them for the extensive revision of the paper. 

Discussion section need to be presented with avoid of repetition of the results. Moreover, some of the provided comments sound a bit out of the context. Authors need to rewrite the discussion section wit help of the more experience colleagues and clearlly link their observations with appropriate examples from the literature in a logic way.

Please, pay additional attention to the reference list and correct it according to the recommendations from the journal. Pay attention to use of capitals where needed (Title of the Journal). Journal titles need to be presented in abbreviated form; year of publication need to be in bolt.  

Author Response

Response to editor and reviewers’ comments

Dear Editors and Reviewers,

Thank you for the comments and helpful suggestions, which are very helpful for revising and improving our paper, as well as the important guiding significance to our further research. We have studied comments carefully and have revised this manuscript accordingly. Changes are highlighted in yellow, and responses to their specific comments are detailed below.

Sincerely yours,

Dr. Fengwei Tian

State Key Laboratory of Food Science and Technology

Jiangnan University

Wuxi 214122, P. R. China

fwtian@jiangnan.edu.cn

Q1: Title is very long and can be presented in better, more compact way.

Response: Thank you for your comments. We changed the title as “Ganoderma lucidum ethanol extraction promote DSS-induced colitis recovery and modulation in microbiota”.

Q2:Please, decide if you can call your country PR China or China. Personally, I would like to follow names according to the United Nation List.

Response: Thank you for your comments.  The People's Republic of China was the full name to the United Nation List, and the P. R. China is the abbreviation of it.

 Q3: In order to be called probiotic, a microorganism supposed to be provided as life and in certain numbers to the host where can exhibits his beneficial properties. Associated with the definition, edible mushroom will be difficult to be defined as probiotics. Maybe different, more appropriate way of calling them need to be choice by the authors. If authors using different extracts from these mushrooms, then this extract can be considered as prebiotics, but this is different term, referring to the nondigestive organic structure that can be associated with the probiotics (microorganisms) growth. However, this need to be well defined by the authors.

Ln14-18: All information presented on this section is showing well, that authors exploring PREBIOTIC role of the GLP, GIP, GLE, GIE on the specific bacterial species presented in the GIT or direct effect of those extracts and consequence effecting the colon inflammation. However, this is not a PROBIOTIC effect. Please, check what is probiotic, prebiotic.

Response: Thank you for your reminder. We had re-defined the edible mushroom as prebiotic in full text, as in line 18, line 19, line 48, line 49, line 53, line 65, line 149, line 151, line 163, line 169, line 202, line 291 and line 358.

 Q4: Ln45: Provide references about polyphenols from mushrooms; Ln58-62 can be combined and avoid repetitions; Introduction need to be prepared according to the specific meaning of the terms. As well, information in the introduction needs to be clearly associated with the feature presented information explored by the article. Please, reorganize and present Introduction in better way.

Response: Thank you for your comments. We reorganized the Introduction and modified according to your comments in line 52-64, as “Prebiotics showed alleviative effect on IBD as maintenance the balanced intestinal flora, enhancement intestinal barrier function, promoting intestinal immune tolerance, interfering with intestinal inflammation, and inhibiting apoptosis of intestinal epithelial cells [10]. Edible mushrooms Ganoderma lucidum and Gloeostereum incarnatum as prebiotics used in traditional Chinese medicines to treat various diseases and to promote physical well-being [11-14]. G. incarnatum extracts possess antioxidant, immunomodulatory, anti-inflammatory, antibacterial, and antitumor (anticancer) activities [15-17]. G. lucidum polysaccharides (GLPs) include (1→3), (1→6)-a/β-glucans, glycoproteins showed antihypoglycemic, antitumor (anticancer), anti-fatigue, immunomodulatory, antioxidant, antihypolipidemic, anti-inflammatory, and anti-decrepitude (prolonging life) activities [18-21]. Ganoderma lucidum ethanol extract including ganoderic acids and saponins ameliorated lipid metabolic disorders, hypoglycemic, antioxidant and modulates the gut microbiota [22-25]”.

 Q5: Ln81: Add country, presume that this is China.

Response: Thank you for your reminder. We had added the country in line 74 as “G. lucidum and G. incarnatum were obtained in October from the Jilin, P. R. China and the Yunnan, P. R. China, respectively”.

 Q6: Ln78-86: Process of preparation of the extract need to be presented better with sufficient details in order experiments to be repeated. Moreover, it is well known that season of collection of medical plants, including mushrooms is important for the presence of some components. Thus, period of collection of mushrooms and their keeping/treatment as well need to be provided. Way of remove of proteins, as well need to be described, and appropriate reference to be provided.

Response: Thank you for your comments. We added sufficient details, season of collection of medical plants, treatment, and the reference of remove of proteins in line 74-78 as “G. lucidum and G. incarnatum were obtained in October from the Jilin, P. R. China and the Yunnan, P. R. China, respectively. Obtained G. lucidum and G. incarnatum were dried in 50℃. G. lucidum water extraction polysaccharides (GLP) and G. incarnatum water extraction polysaccharides (GIP) were extracted through 95℃ hot water bath for 30 minutes 3 times and then 75% ethanol precipitation”, and reference 27 line 82.

 Q7: Most of the material and methods were described in a very poor way, missing details, missing references.  

Response: Thank you for your comments. We supplemented the material and methods such as how to calculate the weight changes (%) as “weight changes (%) performance as measured body weight (day 8-14)/ body weight (day 0)” in line 93-94, scoring system for Disease Activity Index (DAI) in Table 1 line 95, ELISA measurement as “Frozen colon tissues were suspended with PBS at 1:9 ratio. After centrifugation at 12,000 rpm 15 min, collected the supernatant for activity measurements. MUC2, tight junction (ZO-1 and Occludin), COX-2, iNOS, and MPO were measured using ELISA kits (Nanjing SenBeiJia Biotechnology Co., Ltd., Nanjing, Jiangsu, P. R. China) according to the manufacturer’s instructions and as previously reported [29] ” in line 139.

Table 1. Scoring system for Disease Activity Index (DAI)

Score

Weight loss

Stool consistency

Blood stool

0

No loss

Normal

No blood

1

1-5%

Loose stool

2

5-10%

Watery diarrhea

Presence of blood

3

10-20%

Slimy diarrhea, little blood

4

>20%

Severe watery diarrhea with blood

Gross bleeding

 Q8: Results are presented more-or-less in a good shape, however, most probably based on unexperienced position of the authors, in several places wrong terminology was used. Maybe authors can look for a help from more experience colleagues to help them for the extensive revision of the paper. 

Response: Thank you for your comments. We reorganized the Results modified according to your comments. As 3.1 in line 148-161 “DSS-induced colitis in mouse model was used to evaluate whether G. lucidum and G. incarnatum extracts had an obviously prebiotic effect on IBD.  DSS-induced colitis solution shown in Figure 1A, after analysis we found GLE administration at 50 and 100 mg/kg showed better prebiotic effect in promote recovery from colitis, displayed as significantly decreased weight loss (Figure 1B), reduced mortality (Figure 1C), reduced weight loss (Figure 1B), significantly decreased colon shortening (Figure 1D), and lower DAI scores was observed (Figure 1E). Compared with the recover group GLP administration in 500 mg/kg didn’t showed effectiveness in promoting recovery in decreased weight loss (Figure 1B), reduced mortality (Figure 1C), reduced weight loss (Figure 1B), decreased colon shortening (Figure 1D), and lower DAI scores (Figure 1E).  Compared with GLP administration in 500 mg/kg, 500 mg/kg GIP and 100 mg/kg GIE administration promoted weight gain while showed no difference in reduced mortality (Figure 1C), reduced weight loss (Figure 1B), decreased colon shortening (Figure 1D), and lower DAI scores (Figure 1E)”, 3.2 in line 169-185 and line 187-194, “To further evaluate its prebiotic effect of GLP, GLE, GIP, and GIE against DSS-induced colitis in mice, histological analyses were performed by staining with hematoxylin and eosin (H&E). In the recovery, GLP, GIP, and GIE groups, locally seen more epithelial cells and incompletely structure of the mucosal layer, connective tissue hyperplasia with a small amount of lymphocytic infiltration and more focal lymphocytic infiltration in the lamina propria (Figure 2A). Low histological scores epithelial erosion scores, goblet cell depletion scores, crypt damage scores, mucosal edema scores and inflammatory infiltration scores were shown in 50 and 100 mg/kg GLE groups, and the lower scores were in 100 mg/kg GLE (p < 0.05) (Figure 2B-G). In 50 and 100 mg/kg GLE groups, the mucosal layer was structurally intact, with closely arranged epithelial cells and abundant glands in the lamina propria, and a small local focal infiltration of lymphocytes (Figure 2A). Compared with the recovery group, the GLP and GIE groups showed no significant difference in the histological score, and the epithelial erosion, goblet cell depletion, crypt damage scores, mucosal edema and inflammatory infiltration showed no significant difference (Figure 2B-G). The GIP-treated exhibited a lower mucosal edema score than the recovery group, and other scores revealed a trend consistent (Figure 2B-G)”, “No significant difference was seen in MPO concentrations at control, DSS, recover, GLP, GLE, GIP, and GIE groups (Figure 3B). Compared with the recover group, GLP administration at 500 mg/Kg, GLE administration at 50 and 100 mg/kg significantly decreased the concentrations of iNOS and COX-2 (Figure 3A, C). The concentrations of iNOS and COX-2 in GLE administration were lower than in GLP administration (Figure 3A, C). Compared with the recover group, GIE groups showed no difference in the concentrations of iNOS and COX-2 (Figure 3A, C)”, 3.4 in line 223-235 as “The concentration of mucin 2 (MUC2), claudin-3, occludin ZO-1, and were measured to analysis the effects of G. lucidum and G. incarnatum extracts for intestinal barrier protection. Firstly, DSS indeed destructed the intestinal barrier as significant decreased the concentrations of MUC2, ZO-1, Claudin-3 and Occludin (Figure 5). During the recovery period intestinal barrier showed repairing effect as increased concentrations of ZO-1 and Claudin-3, while there were no different in concentrations of MUC2 and Occludin (Figure 5). The concentrations of MUC2, ZO-1, Claudin-3 and Occludin in GLE administration showed no significant difference with control group, indicating that GLE administration promoted the reparation of the intestinal barrier and showed no difference with control group (Figure 5). The concentrations of MUC2, ZO-1, Claudin-3 and Occludin in GLP, GIP and GIE administration showed no different in recover, demonstrated that GLP, GIP and GIE could not promote the recovery in intestinal barrier (Figure 5)”, 3.5 in line 256-261 as “Heat map at the genus level displayed, compared with the recover group, 100 mg/kg GLE significantly increased the abundance of Turicibacter and Bifidobacterium and decreased the abundance of Bacteroides, Staphylococcus, and Romboutsia; 50 mg/kg GLE significantly increased the abundance of Enterococcus and Parabacteroides and decreased the abundance of Staphylococcus, and Romboutsia (Figure 6E)”.

 Q9: Discussion section need to be presented with avoid of repetition of the results. Moreover, some of the provided comments sound a bit out of the context. Authors need to rewrite the discussion section with help of the more experience colleagues and clearly link their observations with appropriate examples from the literature in a logic way.

Response: Thank you for your comments. In order to avoid the repetition of the results we reorganized the Discussion modified according to your comments in line 288-295 as “G. lucidum ethanol extract was enriched with triterpenoids including ganoderic acids and saponins, which has wide pharmacological effects, such as anti-obesity, increase immunity and anti-inflammatory activities [23,32,33]. In addition, our study certificated G. lucidum ethanol extract showed a prebiotic effect on DSS-induced colitis. Although Guo et al. demonstrated that the application of GLP prevented DSS-induced colitis in mice by enhancing intestinal barrier function, increasing the production of SCFAs, and regulating the intestinal microbiota [34]. And in our research GLP did not display promoting effects in the recovery from colitis”, line 296-311 as “G. lucidum ethanol extract increase immunity during the recovery period. G. lucidum ethanol extract administration significant reduced iNOS and COX-2 activities. high levels of COX-2 are associated with gastroenteritis and unregulated iNOS is frequently noted in chronic inflammation, and these molecules are classical biomarkers for assessing the level of inflammatory response in the colon [36-38]. Also G. lucidum ethanol extract administration decreased the expression of inflammatory mediators IL-1β, IL-6, IL-17, and TNF-α in DSS-induced colitis. The levels of inflammatory factors contribute to the initiation, development, and progression of DSS-induced colitis [35]. Overproduction of pro-inflammatory cytokines, including TNF-α, IL-1β, IL-17, and IL-6, causes severe tissue damage and enhances the inflammatory response [39-42]. Meanwhile, after GLE administration the levels of anti-inflammatory cytokines (IL-10 and PPAR-γ) were consistent with those in the control group. IL-10 and PPAR-γ are immune-modulating cytokines with anti-inflammatory activity, and they have shown the capability to reduce a clinical symptom in patients with IBD [43]. Edible mushroom could enhance immune responses via MAPK and NFκB signal pathways [60], and GLE administration immune responses were in TRAF6/ MyD88/NF-κB signaling pathway”, line 312-316 as “Intestinal inflammation in IBD can disturb the epithelial barrier integrity, causing increased permeability and infiltration of pathogens [44]. GLE administration promoted the recovery of the intestinal barrier through promoted the expression of tight junction and adherens junction. Therefore, after GLE administration, the concentrations of MUC2, ZO-1, claudin-3, and occludin were similar with the control group”, line 324-326 as “GLE-treated mice showed significant clustering separation from DSS-induced mice using PCoA, which suggests that GLE treatment noticeably altered the biological structure” and line 337-341 as “The abundance of Bacteroides is negatively correlated with the extent of tight junction proteins (ZO-1, occluding and claudin-1), and positively associated with levels of pro-inflammatory cytokines (IL-1β, TNF-α and IL-6) [48]. There was a significant increase of Bacteroides abundance during the recovery period, and high-dose GLE administration could reduce the abundance of Bacteroides”. We also re-write the conclusion part in line 358-367 as “In conclusion, we evaluated the prebiotic effects of G. lucidum and G. incarnatum different consumption ways (GLP, GLE, GIP, and GIE) on DSS-induced colitis. GLE demonstrated effective effects in promoting recovery from colitis as prevented weight loss, reduced mortality, relieved colonic shortening, and reduced the DAI and histological scores. In contrast, GLP, GIP, and GIE did not have a positive effect on relieving colitis. Recovery-promoting effect of GLE is mainly achieved by increasing immunity and protecting the intestinal barrier. In addition, GLE administration altered the gut microbiota composition, significantly increased the abundance of Turicibacter, Bifidobacterium, and Parabacteroides and decreasing the abundance of Escherichia_Shigella, Bacteroides, and Staphylococcus

 Q10: Please, pay additional attention to the reference list and correct it according to the recommendations from the journal. Pay attention to use of capitals where needed (Title of the Journal). Journal titles need to be presented in abbreviated form; year of publication need to be in bolt.  

Response: Thank you for your reminder. We re-checked and corrected the reference list according to the recommendations from the journal in line 421-632, such as “Malinowski, B.; WiciÅ„ski, M.; SokoÅ‚owska, M.M.; Hill, N.A.; Szambelan, M. The rundown of dietary supplements and their effects on inflammatory bowel disease—A review. Nutrients 2020, 12, 1423”.

Reviewer 3 Report

Dear authors,

I revised the manuscript with ID foods-1926997, which presents a relevant topic that falls in the scope of the journal. The manuscript is well written, the methods are presented in an appropriate manner, and the results are discussed in depth in correlation with the available literature. Also, the conclusions support the results achieved through the experimental work. Still, there must be mentioned the limitations of the work by being specified what else must be done to achieve better results.

Sincerely,

Author Response

Response to editor and reviewers’ comments

Dear Editors and Reviewers,

Thank you for the comments and helpful suggestions, which are very helpful for revising and improving our paper, as well as the important guiding significance to our further research. We have studied comments carefully and have revised this manuscript accordingly. Changes are highlighted in yellow, and responses to their specific comments are detailed below.

Sincerely yours,

Dr. Fengwei Tian

State Key Laboratory of Food Science and Technology

Jiangnan University

Wuxi 214122, P. R. China

fwtian@jiangnan.edu.cn

I revised the manuscript with ID foods-1926997, which presents a relevant topic that falls in the scope of the journal. The manuscript is well written, the methods are presented in an appropriate manner, and the results are discussed in depth in correlation with the available literature. Also, the conclusions support the results achieved through the experimental work. Still, there must be mentioned the limitations of the work by being specified what else must be done to achieve better results.

Response: Thank you for your comments. Firstly, because the title is very long, we changed it in a better present. Secondly, in order to clearly associated with the feature presented information explored by the article, we reorganized the introduction. Thirdly, because of the missing details and references, we supplemented the material and methods. Fourthly, due to lack of experience, we reorganized the results and modified in a better way. Fifthly, lots of repetition results in discussion section, we reorganize logical relationships and rewrite the discussion section. Sixthly, we corrected the reference according to the recommendations from the journal. Seventhly, re-analyzed the data and re-displayed the figures in Figure1-7. Finally, thank you for the comments and helpful suggestions, which are very helpful for revising and improving our paper, as well as the important guiding significance to our further research.

Round 2

Reviewer 2 Report

In my opinion authors have corrected and improved quaility of the presentation and in present way, paper is acceptable for publication